# Target Attainment and Clinical Efficacy for Vancomycin in Neonates: Systematic Review

**DOI:** 10.3390/antibiotics10040347

**Published:** 2021-03-25

**Authors:** Marta Mejías-Trueba, Marta Alonso-Moreno, Laura Herrera-Hidalgo, Maria Victoria Gil-Navarro

**Affiliations:** 1Unidad de Gestión Clínica de Farmacia, Hospital Universitario Virgen del Rocío, 41013 Seville, Spain; marta.mejias.sspa@juntadeandalucia.es (M.M.-T.); marta.alonso.moreno.sspa@juntadeandalucia.es (M.A.-M.); 2Unidad de Gestión Clínica de Farmacia, Instituto de Biomedicina de Sevilla (IBiS), Hospital Universitario Virgen del Rocío, 41013 Seville, Spain; mariav.gil.sspa@juntadeandalucia.es

**Keywords:** vancomycin, neonates, review, target attainment, clinical efficacy

## Abstract

Vancomycin is commonly used as a treatment for neonatal infections. However, there is a lack of consensus establishing the optimal vancomycin therapeutic regimen and defining the most appropriate PK/PD parameter correlated with the efficacy. A recent guideline recommends AUC–guided therapeutic dosing in treating serious infections in neonates. However, in clinical practice, trough serum concentrations are commonly used as a surrogate PKPD index for AUC24. Despite this, target serum concentrations in a neonatal population remain poorly defined. The objective is to describe the relationship between therapeutic regimens and the achievement of clinical or pharmacokinetic outcomes in the neonatal population. The review was carried out following PRISMA guidelines. A bibliographic search was manually performed for studies published on PubMed and EMBASE. Clinical efficacy and/or target attainment and the safety of vancomycin treatment were evaluated through obtaining serum concentrations. A total of 476 articles were identified, of which 20 met the inclusion criteria. All of them evaluated the target attainment, but only two assessed the clinical efficacy. The enormous variability concerning target serum concentrations is noteworthy, which translates into a difficulty in determining which therapeutic regimen achieves the best results. Moreover, there are few studies that analyze clinical efficacy results obtained after reaching predefined trough serum concentrations, this information being essential for clinical practice.

## 1. Introduction

Vancomycin is a glycopeptide antibiotic commonly used for the treatment of neonatal infections caused by coagulase–negative staphylococci (CoNS), methicillin–resistant *Staphylococcus aureus* (MRSA) and enterococci species [1,2]. Due to the increase in methicillin–resistant staphylococcal strains, vancomycin is often used as an empiric therapy in this population.

However, even though vancomycin is widely used in neonates, there is a lack of consensus establishing the optimal vancomycin therapeutic regimen [2], due to its high pharmacokinetic (PK) variability and greater interindividual variability than in the adult population [3]. The neonatal population is characterized by a higher body water percentage, reduced protein binding and higher free fraction [4] and decreased renal clearance at birth, which gradually increases as the renal system matures [5]. This fact must be taken into consideration, since vancomycin can cause nephrotoxicity, because renal is the primary route of excretion. Therefore, therapeutic drug monitoring (TDM) takes on a fundamental role in these patients.

Another controversial point is the definition of the most appropriate PK/PD parameter correlated with the efficacy of vancomycin treatment. In an adult population, the ratio of the area under the concentration time curve over 24 h to the minimum inhibitory concentration (AUC24/MIC) greater than 400 has been found to be the best predictor of successful treatment against MRSA lower respiratory tract infections [6]. A recent guideline reviewed by a consensus of different scientific societies [7] recommends AUC–guided therapeutic dosing for a successful outcome of treatment for an MRSA infection for all neonates. However, in clinical practice, due to the small amount of blood and the difficulty in estimating the AUC24 in the neonatal population, trough serum concentrations (Cmin) are commonly used as a surrogate PKPD index of AUC24/MIC [8,9].

For example, Neofax [10] pharmacotherapeutic guidelines recommend Cmin as a PK target, with a target range of 5–10 mcg/mL for most infections, and of 15–20 mcg/mL when treating MRSA pneumonia, endocarditis, or bone infections in neonates. Nevertheless, it should be stressed that the target exposure was derived from adult studies, and no study has been conducted to validate this in a neonatal population [11].

The main goal of this systematic review is to describe the relationship between therapeutic regimens proposed for the neonatal population and the achievement of clinical or pharmacokinetic objectives, as well as their safety.

## 2. Results

### 2.1. Bibliographic Search

A total of 476 articles were identified, of which 470 were obtained from the different databases that were consulted (185 from PubMed and 285 from EMBASE), as well as six records identified through reference and citation searches of the included papers. After eliminating duplicates, a total of 359 articles were left. Of these, 117 were removed using Mendeley via duplicate checking. 

A further 304 articles were excluded based on their title and summary. The 55 potentially relevant studies were retrieved in full text, of which 35 were excluded before data extraction, and 20 met the inclusion criteria and were, therefore, included in this systematic review (Figure 1).

### 2.2. Quality of the Included Studies

The methodological quality of the studies included in this review was variable. Zero studies were evaluated as good quality or low risk of bias, 15 studies were assessed as having some concerns or a moderate risk of bias and five studies were reported as poor quality or having a serious risk of bias. The detailed results of the risk of bias assessment are summarized in Table 1.

### 2.3. Characteristic of the Included Studies

As shown in Table 2, of the 20 studies included, only one [31] was a randomized controlled trial. Of the remaining studies, ten were retrospective studies [13,14,17,18,19,20,21,24,25,30], seven were prospective studies [15,16,22,26,27,28,29] and two used a mixed methodology (partly retrospective and partly prospective) [12,23].

These 20 studies describe a total of 30 populations of patients.

Table 2 and Table 3 present the variables and results of the articles included in this review.

### 2.4. Serum Concentrations and Dosage Form

Of the 20 articles identified, 13 administered the vancomycin through an intermittent intravenous infusion, four exclusively used continuous infusion and the remaining three compared a regimen based on continuous and intermittent infusions.

The target serum concentrations that were predefined in the different studies were very disparate; in the case of intermittent infusion, five studies set a trough serum concentration of 5–10 mcg/mL [12,23,26,27,30], two studies set the range of 10–15 mcg/mL [20,24], eight set it between 10 and 20 mcl/mL [13,14,18,19,21,25,28,31] and one between 10 and 25 mcg/mL [29]. Regarding the target concentrations in the case of continuous infusion, four established a range of 15–25 mcg/mL [15,22,28,31], one of 15–20 mcg/mL [30], another of 10–30 mcg/mL [16] and the last of 20–30 mcg/mL [17].

### 2.5. Dosage Regimen Used

The main variables, which were based on the different dosing regimens used in the 30 populations of patients that were included, were collected. It should be noted that some regimens take different factors into account. Hence, six regimens took into account the patients’ creatinine levels; six included weight; and 22 included the patients’ age, whether by calculating gestational, postmenstrual and/or postnatal age.

All the populations included in the seven articles in which vancomycin was administered through a continuous infusion, except for a subgroup of patients in the article by Pawlotsky et al. [16], included the loading dose in the dosage regimen. The dose used in the majority of cases [17,22,28,30,31] was 10 or 15 mg/kg, while in the remainder of cases, the dose was 7 mg/kg [16]. In the article by Leroux et al. [15], this loading dose was calculated using a predefined formula (target concentration per volume of distribution).

### 2.6. Main Findings

#### 2.6.1. Efficacy

Two of the articles included in this review evaluated the clinical efficacy of the vancomycin treatment [29,30]. One of these analyzed the bacteriological efficacy, defined as the negativization of cultures 48 and 96 hours after the start of the vancomycin treatment [29], whereas the second article also evaluated the negativization of the cultures 48 h after the start of the antibiotic, as well as the failure of the treatment [30], defined as death from the infection or deterioration of clinical, laboratory and radiological statuses, despite the treatment, and the Töllner score [32]. Regarding the analysis of the variable “negativization of cultures”, the first of the articles [29] found that in 71.3% (*n* = 57) of patients, the cultures with CoNS isolates were negative 48 h after the start of the treatment, with this figure being 93% (*n* = 76) after the end of the antimicrobial treatment. Meanwhile, in the study published by Demirel et al. [3], the subgroup of patients who received vancomycin intermittently presented a negativization of 57.9% of the cultures 48 h after the start of the antimicrobial treatment, and of 63.6% of cultures in the subgroup that received vancomycin through a continuous infusion.

Concerning the target attainment, all identified studies analyzed this factor, with widely different results with regard to the percentage of patients that achieved the target serum concentration. The percentage of neonates who reached the serum concentrations defined by the authors through receiving intermittent vancomycin was very variable (range: 4%–75%), and only 12.5% of the dosage regimens including intermittent vancomycin led to more than 70% of the patients being within the therapeutic range [29]. With regard to the studies which opted for a continuous infusion, the percentage of patients who achieved the serum concentrations varied between 41% and 88%; while 50% of the regimens [15,16,22] led to 70% of neonates being within the range.

Three articles [28,30,31] that compared the continuous and intermittent infusions obtained better initial pharmacokinetic results (in the first few hours) by using continuous vancomycin (range: 53%–85%) compared to intermittent infusion (range: 34%–46%).

#### 2.6.2. Safety

Eleven of the articles evaluated safety, of which ten studied the nephrotoxicity [13,15,16,21,25,26,27,28,30,31] and another exclusively evaluated the variation in creatinine levels (after 48 h, compared to the base levels) [29]. Safety was not analyzed in the nine remaining articles.

Of the ten articles that included the possible appearance of nephrotoxicity, only two of them noted such adverse effect [21,25] in 6.9% to 8.3% of patients.

## 3. Discussion

Focusing on this review, only two papers dealt with the clinical efficacy of vancomycin [29,30]. In relation to clinical efficacy, the study carried out by Plan et al. [29], where vancomycin was administered by intermittent perfusion, revealed that, at 48 h, 71.3% of the CoNS cultures were negative and showed a negativization of the cultures of 93% by the end of the treatment. However, the study carried out by Demirel et al. [30], which compared intermittent versus continuous administration, found that, at 48 h, 57.9% of cultures became negative in the intermittent infusion group and 63.6% in the cohort of continuous infusion.

Among the rest of the studies, only six dosage regimens, depicted in seven articles, achieved more than 70% of patients reaching the serum concentrations established by the authors, of which two were administered through intermittent infusion and the remaining four through continuous infusion. The two regimens based on intermittent infusion took into account weight and the creatinine values at the time of dosing the patients, with a dose fluctuating between 15 and 30 mg/kg/day, with the target range set between 10 and 25 mcg/mL [29]. Of the four regimens that used continuous infusion, all of them [15,16,22,28,31] took the weight (fixed dosage per kg of weight) and age of the patient (three of which the postmenstrual age (PMA) and one the postnatal age (PNA)) into account, and in two of them, also creatinine [15,28,31]. In this case, the established target therapeutic range was broad, ranging from 12 to 30 mcg/mL. There was a difference in target serum concentrations based on the method of administering vancomycin (through intermittent or continuous infusion), since the PK profiles of both are very different. In the case of intermittent infusion, target serum concentrations were set with a higher range of fluctuation (5–25 mcg/mL) compared to those used in the case of continuous infusion, for which the values also varied, but to a lesser extent (15–30 mcg/mL). The issue is that, regardless of the way in which vancomycin is administered, there is variability when it comes to defining the values responsible for target attainment. Notably, the differences in the target concentrations used for intermittent regimes are also due to changes in perception over the years; thus, four of the five studies carried out between 1995 and 2011 set target levels between 5 and 10 mcg/mL, while those carried out from 2014 fundamentally established concentrations between 15 and 20 mcg/mL.

When the two available forms of administration are compared, continuous infusion seems to achieve better results in newborns, as well as being much more straight forward to relate to a target. In addition to this, and as reflected in the systematic review carried out by Gwee et al. [33], studies performed in adult populations have shown better clinical and pharmacokinetic results in the achievement of target serum concentrations by using continuous infusion, which also appears to be the case in the neonatal population. This is reflected in that the three studies [28,30,31] that compare the results obtained by using continuous infusion as opposed to intermittent infusion favor the former, as there is a higher percentage of patients who achieve serum concentrations by using this form of dosage. However, the study carried out by Demirel et al. [30] did not achieve at least 70% of the patients reaching the target serum concentration through continuous infusion. This can be explained by the fact that, although a subgroup of the study was given vancomycin through a continuous infusion, the dosage of the drug was calculated as the sum of the total daily doses of the regimen used in intermittent infusion, which means that the patients received a smaller dose of vancomycin than in the rest of the studies that used continuous infusion.

It is notable that regimens of continuous infusion used in the studies of Patel et al. [28] and Gwee et al. [31] are identical, achieving 82% and 85% of patients in the range, respectively, and without any sign of toxicity. The authors of this review consider that, given that the study carried out by Gwee et al. [31] is a clinical trial, which translates into more solid scientific evidence, and its simplicity and the favorable pharmacokinetic results obtained in both studies, this regimen could be a favorable option for the neonatal population, although it would be convenient to have data on clinical and microbiological efficacy in order to corroborate this.

As most of the studies did not analyze efficacy variables, we do not know if the treatment failed in the populations that did not achieve the pharmacokinetic target. It is essential that these serum concentrations translate to a favorable clinical evolution for patients, with the drug serum concentrations being mere surrogate variables for the optimal indicator of the treatment. Therefore, it is necessary to consider different aspects.

The first aspect to consider is the focus of infection, as deep–seated foci require greater exposure to the drug in order to ensure adequate tissue penetration. However, in neonatal populations, isolated microorganisms can sometimes be less invasive than MRSA or than less deep–seated foci of infection, requiring lower vancomycin serum concentrations to achieve the PK/PD target [19,34]. These factors, among others, contribute to a large disparity in target therapeutic ranges established by the different authors included in this review, which translates into the need to reach a consensus and establish which is the most appropriate.

Another aspect to determine which dosage regimen achieves better results is the fact that these results must be comparable, so the trough serum concentrations should be equal or at least similar. Moreover, these Cmin levels are still only surrogate values for PK/PD efficacy. This, together with the aforementioned difficulty in measuring the main variable, greatly hinders the establishment of PK/PD parameters in clinical practice for this population. Despite this, applications based on modeling and simulation have recently been made available to help determine AUC and facilitate clinical decisions [35].

Likewise, it is essential to identify the dosage regimen that is closest to the patients’ needs and that leads to improved target serum concentrations, but most importantly to a favorable clinical response to the treatment. The work carried out by many studies has analyzed the PK characteristics of vancomycin in neonates, with the aim of identifying the main variables that should be considered when deciding dosages for this population [36,37,38,39]. Therefore, it is crucial to identify and use the dosage regime involved. Other authors also concur that renal function [37,38,39] and postmenstrual age [39] are key factors to consider. A case in point is the study carried out by Hoog et al. [36], who analyzed the different pharmacokinetic determinants in a neonatal population, concluding that renal function and postmenstrual age are the primary factors.

Two of the aforementioned regimens [15,22] calculated the total dose that a patient would individually receive, using a formula composed of different pharmacokinetic variables. Although both obtained that 72% and 70.7% of patients reach serum concentrations in this range, this dosage method is the most difficult to extrapolate to clinical practice, compared to the others [16,28,31], which obtained a higher number of patients in the range using simpler dosages.

Regarding the adverse effects related to the use of vancomycin in the neonatal population, only two studies [21,25] notified the appearance of nephrotoxicity in a small percentage of patients (8.3% and 6.9%, respectively), which was reversible in every case. This indicates that the tested dosing regimens seem to be a safe option for the treatment of Gram–positive infections in this population.

### Strengths and Limitations of the Study

The strength of this review lies in showing the great variability that exists both in establishing adequate dosage regimens as well as in setting the most suitable target serum concentrations for this population. This wide variability makes it difficult to determine exactly which dosage enables the greatest clinical and target attainment as well as the lowest toxicity.

Among the limits of this review, it is essential to highlight that, of the 20 articles included in this review, only one of them is a clinical trial, the rest being observational studies. This is partly understandable, given that the population that concerns us is neonatal patients, with their corresponding vulnerability and difficulty in carrying out clinical trials. It should also be mentioned that five of the articles included present a severe overall bias and the rest moderate, so the results obtained must be evaluated taking this limitation into account. Another limitation to take into account is that this review only considers studies that included vancomycin serum concentrations for patients undergoing treatment, and not in the case of prophylaxis. Furthermore, there was a delay between the end date of the search for articles (April 2020) and the date of its publication, which is due to the large number of studies that were reviewed.

## 4. Materials and Methods

A systematic review was carried out according to Preferred Reporting Items for Systematic Reviews and Meta–Analyses (PRISMA) guidelines [40].

### 4.1. Selection Criteria

Inclusion criteria were defined according to the Population, Intervention, Comparison, Outcome and Study design (PICOS) process, and were as follows:Population: neonatal and young–infant patients (from birth to three months old = 12 weeks) receiving empiric or directed vancomycin therapy.Intervention: monitoring of vancomycin serum concentrations.Comparison: with a comparator (age range, regimen, etc.) or without a comparator.Outcomes: clinical efficacy and/or target attainment, the latter defined as reaching target serum concentrations. Safety of vancomycin treatment through obtaining serum concentrations in those studies that are available.Study design: clinical trials and observational studies.

All articles that did not meet the inclusion criteria were excluded, along with those that used vancomycin as a prophylaxis, those with fewer than 20 serum concentration determinations, those whose populations were patients undergoing renal replacement therapy (hemodialysis, ECMO, etc.) and articles that were written in languages other than English or Spanish.

### 4.2. Data Sources

The bibliographic search was conducted using controlled vocabulary in the literature published from inception until 7 April 2020, in two databases: MEDLINE (through the PubMed interface) and EMBASE.

A search strategy was defined based on the proposed PICOS question and is detailed in Table 4.

To complete the search, articles of interest identified by citation tracing were included.

### 4.3. Study Selection

Firstly, duplicate articles were eliminated. Thereafter, two reviewers (reviewer 1 and reviewer 2) independently selected the articles using the aforementioned inclusion criteria, based on the information obtained from the title and abstract. When in doubt, they read the entire article before deciding whether to include it.

To ensure reproducibility and minimize bias, a third reviewer (reviewer 3) resolved any disagreement. A critical reading of the complete selected articles was then carried out.

### 4.4. Quality Assessment

To evaluate the quality of the studies selected for inclusion, two tools were used according to the study design: the Cochrane Risk of Bias tool for randomized trials (RoB 2) was used for randomized controlled trials [41], and the Risk Of Bias In Non–randomized Studies of Interventions tool (ROBINS–I) was used for non–randomized studies [42]. When using the ROBINS–I tool, the overall risk of bias of the paper was categorized as “Low”, “Moderate”, “Serious” or “Critical”. When RoB 2 was applied, risk of bias was classified as “Low”, “High” or “Some concerns”.

### 4.5. Data Extraction

Reviewer 1 independently extracted data, and reviewer 3 examined all extraction sheets to ensure their accuracy.

A descriptive analysis of the main characteristics of the included studies was carried out, in which different variables were extracted and presented in tables.

The design and target population of each of the identified studies were compiled in Table 2, including the following: author and year of publication, main objective of the study, design, main variable of interest, defined target serum concentrations, method of administration and target population, in which the number of subjects and their demographic characteristics (sex, age, weight, creatinine levels and isolates found) were included.

Furthermore, Table 3 summarizes the information regarding the dosage regimens used in each article (variables making up the dosage regimens, dosages used and dosing intervals), as the main results obtained by the authors. These results were grouped into efficacy, both clinical and target attainment, the former being defined as the infection being healed (the resolution of symptoms or the negativization of cultures), and the latter as achieving target vancomycin serum concentrations, and safety, defined as the emergence of adverse effects or the obtaining of serum concentrations higher than those that had been predefined by the authors.

## 5. Conclusions

This review includes the best available evidence on the relationship between clinical and target attainment and serum concentrations.

Given the wide variability present in the design and objective of the studies, there is insufficient evidence to allow us to recommend a therapeutic regimen for vancomycin treatment in the neonatal population based on the clinical and pharmacokinetic results obtained. However, all the findings point to the fact that dosing by continuous infusion would enable the best results to be obtained, and it is necessary to carry out more RCTs to be able to corroborate these statements.

## Figures and Tables

**Figure 1 antibiotics-10-00347-f001:**
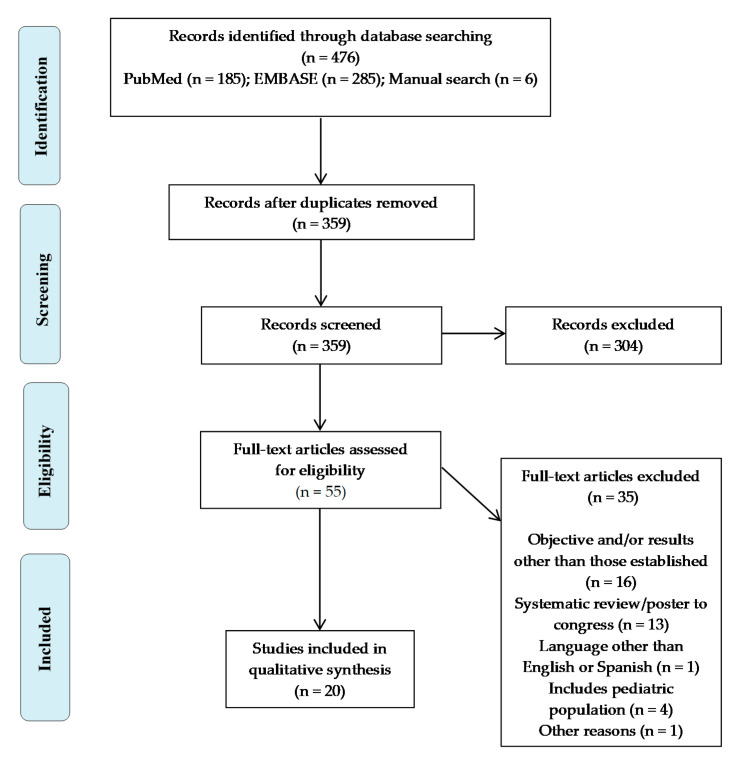
Study selection flowchart.

**Table 1 antibiotics-10-00347-t001:** Quality assessment of the studies.

Study		Risk of Bias Due to							
	Tool	Confounding	Selection of Participants	Classification of Interventions	Deviations from Intended Interventions	Missing Data	Measurement of Outcomes	Selection of the Reported Result	Overall Bias
[12]	ROBINS–I	M	L	L	L	L	M	L	M
[13]		S	L	M	S	L	M	M	S
[14]		S	L	L	L	L	M	M	S
[15]		M	L	L	L	M	M	M	M
[16]		L	L	L	L	L	M	M	M
[17]		M	L	L	L	L	M	M	M
[18]		M	L	L	L	L	M	M	M
[19]		S	L	L	L	L	M	M	S
[20]		S	L	L	L	L	M	M	S
[21]		M	L	L	L	L	M	S	S
[22]		M	L	L	L	M	M	M	M
[23]		M	L	L	L	L	M	M	M
[24]		M	L	L	L	M	L	L	M
[25]		M	L	M	L	M	M	M	M
[26]		M	L	L	L	L	M	M	M
[27]		M	L	L	L	L	M	M	M
[28]		M	L	M	L	L	M	M	M
[29]		M	L	M	L	L	M	M	M
[30]		M	L	M	L	M	M	M	M
**Study**	**Tool**	**Randomization Process**		**Deviations from Intended Interventions**		**Missing Data**	**Measurement of Outcomes**	**Selection of the Reported Result**	**Overall Bias**
[31]	ROB–2	L		SC		L	L	SC	SC

Legend: ROBINS–I: Risk of Bias In Non–randomized Studies of Interventions tool. ROB–2: the Cochrane Risk of Bias tool for randomized trials. L = low; M = moderate; S = serious; SC = come concerns; Y = yes; N = no; NA = not applicable; NR = not reported; NI = not informed. Colors: green = low risk; Yellow = moderate or some concerns risk; Orange = serious risk.

**Table 2 antibiotics-10-00347-t002:** Design and target population of each of the identified studies.

Articles	Main Objective	Design	Main Variable	Target Serum Concentrations and Method of Administration	Target Population					
					Characteristics and No. Subjects	Sex	Age: GE (w) PNA (d) PMA (w)	Current Weight (kg)	Basal Cr (mcmol/L)	Isolates
Aguilar MJ 2008 [12]	Design and validation of an empirical dosing regimen for vancomycin	Retrospective study (regimen design) and prospective validation	No. of neonates and serum concentrations that reach target levels	Intermittent inf: Cmin: 5–10 mcg/mL Cmax: 20–40 mcg/mL	Premature: 1) 53 neonates 2) 30 neonates	1) Male: 43.4% (*n* = 23) 2) Male: 46.7% (*n* = 14)	1) GE 30 ± 3 PNA 23 ± 13 2) GE 31 ± 2 PNA 13 ± 8	1) 1.3 ± 0.5 2) 1.6 ± 0.4	No data	Empirical and targeted therapy
Ringenberg T 2015 [13]	To assess the percentage of neonates and young infants achieving a trough serum concentration	A multi–institutional retrospective chart review	Percentage of NICU patients achieving a trough serum concentration with initial vancomycin dosing	Intermittent inf: 10–20 mcg/mL	141 patients (NICU patients)	Male: 46.1% Female: 53.9%	GE 28.2 ± 4.1 PNA 34.1 ± 34.6 PMA 33.1 ± 6.3	1.602 ± 1.015	No data	Empirical: 58.6% Targeted: 41.4%
Dersch–Mills D 2014 [14]	To assess the performance of an empirical vancomycin dosing regimen in achieving target trough levels	Retrospective, observational study of vancomycin doses, levels and pharmacokinetics	Percentage of neonates with initial pre–vancomycin levels of <10 mcg/mL, 10–20 mcg/mL and >20 mcg/mL	Intermittent inf: 10–20 mcg/mL	153 patients (NICU patients)	No data	1) Preterm (*n* = 171): PNA: 12 (1–102) a) GE ≤29 (62%) b) GE 30–36 (28%) 2)Term (*n* = 20) PNA: 13 (1–70)*1	No data	No data	No data
Leroux S 2016 [15]	To evaluate the clinical utility and safety of a model–based patient–tailored dose of vancomycin in neonates	Prospective study	Percentage of neonates with a first therapeutic drug monitoring vancomycin serum concentration achieving the target window	Continuous inf: 15–25 mcg/mL	191 patients (NICU patients)	No data	GE: 31.1 ± 4.9 PNA: 16.7 ± 21.7	1.755 ± 0.873	48.6 ± 21.8	Empirical and targeted therapy
Pawlotsky F 1998 [16]	To define a new dosage schedule in premature neonates	Prospective study (2 cohorts)	Mean vancomycin serum concentrations observed and percentage of patients attaining target concentrations at steady state in each group	Continuous inf target steady state: 12 mcg/mL Target range: 10–30 mcg/mL	53 patients (NICU patients)	No data	1) GE: 29.2 ± 2.9 PNA: 4.3 ± 3.1 PMA: 33.5 ± 3.7 2) GE: 30.5 ± 3.7 PNA: 3.4 ± 3.5 PMA: 33.9 ± 4.8	1) 1.5 ± 0.3 2) 1.8 ± 0.8	No data	75.5% (*n* = 40) Empirical 24.5% (*n* = 13) targeted therapy
Tauzin M 2019 [17]	To determine the proportion of neonates achieving an optimal therapeutic vancomycin level and which dosing regimen is the most suitable for neonates	Retrospective study	Proportion of neonates reaching the target vancomycin serum concentration	Continuous inf: 20–30 mcg/mL	75 preterm neonates (*n* = 91 therapy episodes)	Male: 57.3% (*n* = 43) Female: 42.7% (*n* = 32)	GE: 27 (26–30.5) PNA: 15 (9–33)	1.23 [0.94–1.79]	(*n* = 68) 52 [26.5–70]	73.6% (*n* = 67) empirical 26.4% (*n* = 24) targeted therapy
Chung E 2018 [18]	To evaluate whether vancomycin dosing from published dosing algorithms correlate with the attainment of target trough concentrations	Retrospective study	Proportion of the first minimum levels within the target therapeutic range, as well as in the subtherapeutic range within therapeutic and subtherapeutic levels	Intermittent inf: 10–20 mcg/mL	74 patients *n* = 97 levelsNICU patients	Male: 58.8% (*n* = 57) Female: 41.2% (*n* = 40)	Therapeutic (*n* = 34): GE: 27.6 ± 3.9 PNA: 22.1 ± 16.3 Subtherapeutic: GE 27.6 ± 3.1 PNA: 31.9±26.4	Therapeutic (*n* = 34): 1.334±4.117 Subtherapeutic: (*n* = 63): 1.563 ± 0.736	mg/dL: Therapeutic: 0.64 ± 0.25 Subtherapeutic: 0.45 ± 0.18	24.3% (*n* = 18) empirical 75.7% (*n* = 56) targeted therapy
Radu L 2018 [19]	To validate the empirical vancomycin dosage regimen in achieving target troughs	Multisite retrospective before–and–after cohort study	Proportion of neonates achieving target trough levels	Intermittent inf: 10–20 mcg/mL	118 patients NICU Patients	No data	EG: 28.4 (26.3–34.3) PNA: 15 (8.0–37.5) PMA: 33.4 (29.1–38.5)	1.814 (0.961)	No data	80.51% (*n* = 95) empirical 19.49% (*n* = 23) targeted therapy
Petrie K 2015 [20]	To determine the initial trough level achievement of neonatal vancomycin given dosing according to the British National Formulary for Children	Retrospective study	Percentage of patients achieving a trough serum concentration with initial vancomycin dosing	Intermittent inf: 10–15 mcg/mL	83 patients	No data	EG: 28 (23+1–41+3) PNA:12 (2–187) PMA: 30 (23–52)	1.12 (0.56–4.7)	42 (17–139)	No data
Reilly AM 2019 [21]	To evaluate the implementation of a new vancomycin dosing guideline in improving trough target attainment	Retrospective study	Percentage of neonates who achieve goal trough concentrations	Intermittent inf: 10–20 mcg/mL	Old guideline: 91 patients; New guideline: 121 patients NICU Patients	No data	Old: PNA: 28 ± 26 PMA: 32 ± 5 New: PNA: 18 ± 14 PMA: 29 ±4	Old: 1.59 ± 0.93 New: 1.10 ± 0.58	mg/dL: 0.56 ± 0.29 0.65 ± 0.34	1) 62.6% (*n* = 57) empirical 37.4% (*n* = 34) targeted therapy 2) 72.72% (*n* = 88) empirical 27.28% (*n* = 33) targeted therapy
Zhao W 2013 [22]	To evaluate the results of vancomycin TDM under three different dosing regimens and to optimize vancomycin therapy	Prospective study: dose optimization multicenter study (three hospitals (1,2,3)) and validation	Percentage ofneonates who achieve goal trough concentrations and concentration range	Continuous inf: 15–25 mcg/mL	a) Dose optimization: 207 samples 116 neonates b) Validation: 58 neonates	a) Male: 50.87% (*n* = 59) Female: 49.13% (*n* = 57) b) Male: 60.34% (*n* = 35) Female: 39.66% (*n* = 23)	a) PNA: 26 ± 25; 17(1,120) PMA: 33.8 ± 5.3; 32.7 (24.4, 49.4) b) PNA: 23 ± 33 11 (1–196)	a) Dose optimization: 1) 1.44 (0.46–5.68) 2) 1.64 (0.53–5.68) 3) 1.99 (0.620–4.50) b) Validation: 1.62 (0.66–3.89)	a) Dose optimization: 1) 46 (5–120) 2) 51 (8–228) 3) 48 (11–180) b) Validation: 45 (10–87)	No data
Matthijs de Hoog 1999 [23]	To incorporate new insights in an up–to–date dosing scheme for neonates of various gestational ages	Retrospective study with prospective validation	Number of patients presenting through and peak levels in the different established plasma ranges	Intermittent inf: Cmin: 5–10 mcg/mL Cmax: 20–40 mcg/mL	PNA < 29 days Retrospective: 108 newborns; Prospective: 22 neonates	No data	Retrospective: GE: 28.9 (24–410 PNA: 14 (3–27) PMA: 31 (26–42) Prospective: GE: 29 (25–42) PNA: 11 (7–21) PMA: 31 (27–43)	Retrospective group: 1.045 (0.51–4.41) Prospective group: 1.16 (0.73–3.42)	No data	Empirical and targeted therapy
Sinkeler FS 2014 [24]	To assess the percentage of therapeutic initial trough serum concentrations and to evaluate the adequacy of the therapeutic range in interrelationship with the observed MIC–values in neonates	Retrospective study	Total number (and %) of cases with trough concentrations below and above the therapeutic range (10–15 mg/L)	Intermittent inf: 10–15 mcg/mL	112 neonates NICU patients	No data	GE: 28 (24–41) PNA: 14 (3–112)	1.04(0.5– 4.31)	No data	Only patients with Gram–positive isolation were included
Madigan T 2015 [25]	To compare vancomycin serum trough concentrations and 24–h area under the serum concentration–versus–time curve (AUC24) among very low–birthweight	Retrospective analysis: before and after implementation of a new vancomycin dosing protocol	Vancomycin trough concentrations and predicted AUC_24_	Intermittent inf: 10–20 mcg/mL	57 preterm < 1.5 kg (NICU patients) Control and intervention group	Control Male: 42.9% (*n* = 12) Female: 57.1% (*n* = 16) Intervention Male: 31% (*n* = 9) Female: 69% (*n* = 20)	Control: GE: 26 (24.0–30.1) PMA: 29.1(25–32.6) Intervention: GE: 25.9 (22.9–31.6) PMA: 28.1 (24–37.3)	Control: 0.94 (0.47–1.47) Intervention: 0.91 (0.49–1.49)	Control: 0.65 (0.2–1.4) Intervention: 0.50 (0.2–1.3)	Positive culture Control: 67.9% (*n* = 19) Intervention: 44.8% (*n* = 13)
Badran EF 2011 [26]	To evaluatethe pharmacokinetic parameters of vancomycinfrom data collected during regular monitoring of its serum concentrations	Prospective study	Percentage of patients reaching the target levels and pharmacokinetic variables in the different cohorts	Intermittent inf: Cmin: 5–10 mcg/mL Cmax: 20–40 mcg/mL	151 neonates (NICU patients): divided into 3 groups	Male: 57% (*n* = 86) Female: 43% (*n* = 65)	1) Group <28 weeks GE: 26.9 ± 0.4 PNA: 14.6 ± 11 2) Group 28–34 GE: 30.3 ± 1.7 PNA: 11.6 ± 7.9 3) Group 34 term GE: 36.7 ± 1.8 PNA: 9.8 ± 5.7	No data	No data	No data
McDougal A 1995 [27]	To estimate the vancomycin pharmacokinetic parameters in a neonatal population and prospectively to evaluate these modified dosage guidelines	Prospective study	Clinical characteristics, pharmacokinetic variables, percentage that reach the target levels	Intermittent inf: Cmin: 5–10 mcg/mL Cmax: 25–35 mcg/mL	44 patients (NICU patients)	No data	PMA: Range (27–44) PNA: Range (2–63)	Range (0.720–3.79)	No data	Empirical and targeted therapy
Patel AD 2013 [28]	Compare a dosing regimen with intermittent vs. continuous infusion	Prospective study: 2 groups	Proportion of patients reaching the target level with the first plasma level	Continuous inf: 15–25 mcg/mL Intermittent inf: 10–20 mcg/mL	1) 60 courses + 60 courses 2) 17 patients: 20 courses	No data	1) Continuous: GE: 29 (24–41) PMA: 36 (26–62) 2) Intermittent: GE: 30 (26–41) PMA: 39 (29–45)	Intermittent: 2.2 (1–4) Continuous: 2.22 (0.62–6.9)	Intermittent: 35 (11–79) Continuous: 33 (15–114)	52.9% (*n* = 9) empirical 47.1% (*n* = 8) targeted therapy
Plan 2008 [29]	To evaluate a simplified dosage schedule for continuous–infusion vancomycin therapy	Prospective study: 2 groups	Percentage of patients reaching target levels and bacteriological data	Intermittent inf: 10–25 mcg/mL	145 premature neonates (<34 weeks)	1) Male: 44% (*n* = 32) 2) Male: 53% (*n* = 38)	1) PNA: 11 (7–18) PMA: 28 (26–29) 2) PNA: 10 (8–15) PMA: 27.5 (26–29)	1) 0.94 (0.795–1.14) 2) 0.87 (0.707–1.17)	1) 70 (60–86) 2) 74 (55–104)	43.45% (*n* = 63) Empirical 56.55% (*n* = 82) targeted: 80 with CoNS
Demirel 2015 [30]	To evaluate microbiological outcomes, clinical response and adverse events of vancomycin when administered via continuous intravenous infusion	Retrospective study (2 cohorts, intermittent or continuous intravenous)	Clinical response and microbiological outcomes; percentage of patients reaching target plasma levels	Intermittent inf: 5–10 mcg/mL; Continuous inf: 15–20 mcg/mL	77 preterm NICU patients (<34 weeks)	1) Male: 68.3% (*n* = 28) 2) Male: 52.8% (*n* = 19)	1) GE: 29.3±2.9 PMA: 9 (4–29) 2) GE: 28.6±2.9 PMA: 11 (4–56)	No data	1) –0.1 (–0.3/–0.05) 2) –0.15 (–0.4/–0.05) *	1) Empirical: 53.7% (*n* = 22) 2) Empirical: 69.4% (*n* = 25)
Gwee A 2019 [31]	To determine if CIV or intermittent infusions of vancomycin better achieves target vancomycin concentrations at the first steady–state level and to compare the frequency of drug–related adverse effects	Multicenter prospective randomized controlled trial: 2 groups	The difference in the proportion ofparticipants achieving target vancomycin levels at their first steady–state level	Continuous inf: 15–25 mcg/mL Intermittent inf: 10–20 mcg/mL	104 patients Intermittent: 51 Continuous: 53	Intermittent: Male: 53% (*n* = 27) Continuous: Male: 47.2% (*n* = 25)	Intermittent: GE: 34.4 ± 5.2 PNA: 23 ± 21 Continuous: GE: 34.0 ± 4.4 PNA: 23 ± 19	Intermittent: 2.503 ± 1.137 Continuous: 2.595 ± 0.970	No data	77.88% (*n* = 81) empirical 22.12% (*n* = 23) targeted therapy

* (2nd creatinine–basal creatinine) mg/dL; *1 population data of patients who meet inclusion criteria (*n* = 191) GE: gestational age; PNA: postnatal age; PMA: postmenstrual age; N/A: not applicable; Cr: creatinine.

**Table 3 antibiotics-10-00347-t003:** Dosage regimen used and main findings of each of the identified studies.

	Dosage Regimen Used			Main Findings		
Articles	Variables Involved	Loading Dose	Maintenance dose	Clinics /Levels in Therapeutic Range	Infra /Supratherapeutic	Security
Aguilar MJ 2008 [12]	Weight and age (PNA) 1) < 1 kg + <15 d 2.1) < 1 kg + >15 d 2.2) > 1 kg + <15 d 3) > 1 kg + >15 d	N/A	1) 10 mg/kg e/12 h 2.1) 15 mg/kg e/12 h 2.2) 15 mg/kg e/12 h 3) 13 mg/kg e/8 h	Validation (*n* = 30) 1) Cmin: 50% and Cmax: 55% 2) Cmin: 62% and Cmax: 75% 3) Cmin: 70% and Cmax: 80% Total: Cmin: 60%; Cmax: 73%	Validation (*n* = 30) 1) Cmin: 50% and Cmax: 45% 2) Cmin: 38% and Cmax: 25% 3) Cmin: 30% and Cmax: 20%	No data
Ringenberg T 2015 [13]	Age (PMA and PNA) PMA ≤ 29 w + PNA 0–14 d PMA ≤ 29 w + PNA >14 d PMA 30–36 w + PNA 0–14 d PMA 30–36 w + PNA >14 d PMA 37–44 w + PNA 0–7 d PMA 37–44 w + PNA >7 dPMA ≥ 45 w + PNA All	N/A	10 mg/kg e/18 h 10 mg/kg e/12 h 10 mg/kg e/12 h 10 mg/kg e/8 h 10 mg/kg e/12 h 10 mg/kg e/8 h 10 mg/kg e/6 h	*n* = 171 10–20 mcg/mL: 25.1% (*n* = 43)	*n* = 171 < 10 mcg/mL: 71.9% > 20 mcg/mL: 2.9%	No nephrotoxicity 2 patients: reversible 50% increase in their creatinine. No other adverse drug reactions.
Dersch–Mills D 2014 [14]	Weight and age (PNA) < 1200 kg + 0–7 d 1200–2000 kg + 0–7 d 2000 kg + 0–7 d < 1200 kg + >7 d 1200–2000 kg + >7 d > 2000 kg + > 7 d	N/A	15 mg/kg e/24 h 15 mg/kg e/18 h 15 mg/kg e/12 h 15 mg/kg e/24 h 15 mg/kg e/12 h 15 mg/kg e/8 h	15% (n = 3) 17% (n = 1) 71% (n = 5) 15% (n = 8) 45% (n = 20) 75% (n = 15) Total: 34% (n = 52)	I: 85%; S: 0% I: 83%; S: 0% I: 29%; S: 0% I: 85%; S: 0% I: 52%; S: 3% I: 20%; S: 5% I: 65%; S: 1%	No data
Leroux S 2016 [15]	Birth weight (g), current weight (g), PNA (days), creatinine (mcmol/L)	Target [ ] × Vd Mean: 11.1 mg/kg	Target × CL × 24 h Mean: 28.3 mg/kg/d	*n* = 91 15–25 mcg/mL: 72% (*n* = 136)	*n* = 191 <10 mcg/mL: 3.1% > 30 mcg/mL: 6.3%	No nephrotoxicity
Pawlotsky F 1998 [16]	Age (PMA) Cohort 1: 25–30 w 31–34 w 35–38 w 39–40 w >41 w Cohort 2: 25–26 w 27–28 w 29–30 w 31–32 w 33–34 w 35–36 w 37–38 w 39–40 w 41–42 w 43–44 w >45 w	Cohort 1: N/A Cohort 2: 7 mg/kg	Cohort 1: 10 mg/kg/day 17 mg/kg/day 20 mg/kg/day 24 mg/kg/day 30 mg/kg/day Cohort 2: 10 mg/kg/day 12 mg/kg/day 15 mg/kg/day 18 mg/kg/day 20 mg/kg/day 23 mg/kg/day 26 mg/kg/day 29 mg/kg/day 31 mg/kg/day 34 mg/kg/day 40 mg/kg/day	Cohort 1: 10–30 mcg/mL: 56% (*n* = 13) Cohort 2: 10–30 mcg/mL: 88% (*n* = 26)	Cohort 1: <10 mcg/mL: 44% >30 mcg/mL: 0% Cohort 2: <10 mcg/mL: 8.6% >30 mcg/mL: 3.4%	No cases of hypotension, flushing, red man syndrome. One patient: reversible creatinine increase
Tauzin M 2019 [17]	N/A	15 mg/kg	30 mg/kg/d	*n* = 91 20–30 mcg/mL: 30.8% (*n* = 28): GA < 28 *n* = 17; GA ≥ 28 *n* = 12 PNA ≤ 14 d *n* = 17; >14 d *n* = 12 ≤1 kg *n* = 11; > 1 kg *n* = 18	*n* = 91 <20 mg/L: 44% >30 mcg/mL: 25.3%	No data
Chung E 2018 [18]	Age (PMA and PNA) PMA ≤29 w + PNA 0–14 d PMA ≤29 w + PNA >14 d PMA 30–36 w + PNA 0–14 d PMA 30–36 w + PNA >14 d PMA 37–44 w + PNA 0–7 d PMA 37–44 w + PNA >7 d PMA ≥ 45 w + PNA All	N/A	10 to 15 mg/kg e/18 h 10 to 15 mg/kg e/12 h 10 to 15 mg/kg e/12 h 10 to 15 mg/kg e/8 h 10 to 15 mg/kg e/12 h 10 to 15 mg/kg e/8 h 10 to 15 mg/kg e/6 h	*n* = 85 10–20 mcg/mL: 60.7% (*n* = 52)	*n* = 85 <10 mcg/mL: 39.3%	No data
Radu L 2018 [19]	Age (PMA and PNA) PMA ≤29 w + PNA 0–21 d PMA ≤29 w + PNA >21 d PMA 30–36 w + PNA 0–14 d PMA 30–36 w + PNA >14 d PMA 37–44 w + PNA 0–7 d PMA 37–44 w + PNA >7 dPMA ≥ 45 w + PNA All	N/A	15 mg/kg e/18 h 15 mg/kg e/12 h 15 mg/kg e/12 h 15 mg/kg e/8 h 15 mg/kg e/12 h 15 mg/kg e/8 h 15 mg/kg e/6 h	38.71% (*n* = 12) 50% (*n* = 2) 78.57% (*n* = 11) 68.97% (*n* = 20) 41.67% (*n* = 5) 46.43% (*n* = 13) N/A (*n* = 0) Total: 53.4% (*n* = 63)	I: 61.25%; S: 0% I: 50%; S: 0% I: 21.43% I:10.3%; S:20.7% I: 50%; S: 8.3% I:39.3%; S:14.3% N/A	No data
Petrie K 2015 [20]	Age (PMA) <29 w 29–35 w >35 w	N/A	15 mg/kg e/24 h 15 mg/kg e/12 h 15 mg/kg e/8 h	Level 10–15 mcg/mL: 13% (*n* = 11)	< 10 mcg/mL: 81% > 15 mcg/mL: 6%	No data
Reilly AM 2019 [21]	Age (PMA and PNA) Old PMA <28 w + PNA 0–14 d PMA < 28 w + PNA >14 d PMA 28–33 w + PNA 0–14 d PMA 28–33 w + PNA >14 d PMA 34–37 w + PNA 0–7 d PMA 34–37 w + PNA >7 d PMA >37 w + PNA 0–7 d PMA >37 w + PNA >7 d New PMA <28 w + PNA 0–14 d PMA < 28 w + PNA >14 d PMA 28–33 w + PNA 0–14 d PMA 28–33 w + PNA >14 d PMA 34–37 w + PNA 0–7 d PMA 34–37 w + PNA >7 d PMA >37 w + PNA 0–7 d PMA >37 w + PNA >7 d	N/A	Old 15–20 mg/kg e/24 h 15 mg/kg e/18 h 15 mg/kg e/18 h 10–15 mg/kg e/8–12 h 10 mg/kg e/12 h 10 mg/kg e/8 h 10 mg/kg e/12 h 10 mg/kg e/12 h New 12.5 mg/kg e/12 h 12.5 mg/kg e/8 h 12.5 mg/kg e/8 h 10 mg/kg e/6 h 12.5 mg/kg e/8 h 12.5 mg/kg e/6 h 15 mg/kg e/8 h 15 mg/kg e/6 h	All: 28.6% (*n* = 26) 10% (*n* = 1) 25% (*n* = 2) 17% (*n* = 2) 47% (*n* = 17) 0% (*n* = 0) 25% (*n* = 2) 0% (*n* = 0) 20% (*n* = 2) All: 62% (*n* = 75) 64% (*n* = 21) 40% (*n* = 4) 45% (*n* = 10) 74% (*n* = 29) 0% (*n* = 0) 80% (*n* = 8) 50% (*n* = 1) 50% (*n* = 2)	I: 69.2% (*n* = 63); S: 2.2% (*n* = 2) I: 90%; S: 0% I: 75%; S: 0% I: 83%; S: 0% I: 47%; S: 6% I: 100%; S: 0% I: 75%; S: 0% I: 100; S: 0% I: 80%; S: 0% I: 9% (*n* = 11); S: 29% (*n* = 35) I: 15%; S: 21% I: 10%; S: 50% I: 9%; S: 45% I: 8%; S: 18% I: 0%; S: 100% I: 0%; S: 20% I: 0%; S: 50% I: 0%; S: 50%	Old guideline: Nephrotoxicity: 7.7% New guideline: Nephrotoxicity: 8.3%. No differences were observed between groups
Zhao W 2013 [22]	Age (GA and PNA) GA ≥24<27 w + ≤7 d GA ≥24<27 w + >7 d GA ≥27<30 w + ≤7 d GA ≥27<30 w + >7 d GA ≥30<32 w + ≤7 d GA ≥30<32 w + >7 d GA ≥32 w + ≤7 d GA ≥32 w + >7 d 2) *n*/A 3) N/A	a) 1)10 10 10 10 15 15 15 15 2)15 3) No b) Validation Loading: Target × Vd	NRF: 20 IRF: 15 NRF: 20 IRF: 15 NRF: 25 IRF: 20 NRF: 25 IRF: 20 NRF: 30 IRF: 25 NRF: 25 IRF: 20 NRF: 30 IRF: 25 NRF: 30 IRF: 25 NRF: 35 IRF: 30 NRF: 30 NRF: 30 IRF: 20 b) Calculated based on variables (individualized)	a) Dose optimization: the results broken down by hospitals are not provided: Total 15–25 mcg/mL: 41.4% (*n* = 48 b) Validation: 15–25 mcg/mL: 70.7% (*n* = 41)	a) Dose optimization: The results broken down by hospitals are not provided: <15 mcg/mL: 34% (*n* = 40) >25 mcg/mL: 24% (*n* = 28) b) Validation: < 15 mcg/mL: 15.5% (*n* = 9) > 25 mcg/mL: 13.8% (*n* = 8)	No data
Matthijs de Hoog 1999 [23]	N/A	N/A	Retrospective: 15 mg/kg e/12 h Prospective: 10 mg/kg e/8 h	*n* = 108 T: 5–15 mcg/mL: 65.7% (*n* = 71) P: 20–40 mcg/mL: 77.8% (*n* = 84) *n* = 22: before the 5th dose T: 5–15 mcg/mL: 77.3% (*n* = 17) P: 20–40 mcg/mL: 86.4% (*n* = 19)	*n* = 108 T: <5 mcg/mL: 17.6% >15 mcg/mL: 16.7% P: <20 mcg/mL: 5.6% >40 mcg/mL: 16.7% *n* = 22: before the 5th dose T < 5 mcg/mL: 4.5% >15 mcg/mL: 18.2% P: <20 mcg/mL: 13.6% >40 mcg/mL: 0%	No data
Sinkeler FS 2014 [24]	Age (GA and/or PMA and/or PNA) PMA <26 w GA 26–37 w + PNA <7 d GA >37 w + PNA <7 d PNA >7 d	N/A	15 mg/kg e/24 h 10 mg/kg e/12 h 15 mg/kg e/12 h 20 mg/kg e/12 h	*n* = 112 10–15 mcg/mL: 33.04% (*n* = 37)	*n* = 112 <10 mcg/mL: 47.32% >15 mcg/mL: 19.64%	No data
Madigan T 2015 [25]	Control: unknown Intervention: weight and age (PNA) < 1.3 kg + <7 d <1.3 kg + ≥ 7 d ≥1.3 kg + <7 d ≥1.3 kg + ≥ 7 d	N/A N/A	Control: unknown Intervention: 15 mg/kg e/24 h 15 mg/kg e/12 h 15 mg/kg e/18 h 15 mg/kg e/8 h	Control: 10–20 mcg/mL: 4% (*n* = 1) Intervention: 10–20 mcg/mL: 34% (*n* = 10)	Control: <5 mcg/mL: 50% 5–10 mcg/mL: 46% > 20 mcg/mL: 0% Intervention: <5 mcg/mL: 24% 5–10 mcg/mL: 34% > 20 mcg/mL: 7%	Nephrotoxicity: 2 patients intervention group, 0 patients in control group. Failure hearing: Intervention group: 3/24 Control group 3/22
Badran EF 2011 [26]	Age (PMA and PNA) PMA ≤29 w + PNA 0–14 d PMA ≤29 w + PNA >14 d PMA 30–36 w + PNA 0–14 d PMA 30–36 w + PNA >14 d PMA 37–44 w + PNA 0–7 d PMA 37–44 w + PNA >7 d PMA ≥ 45 w + PNA All	N/A	10 mg/kg e/18 h 10 mg/kg e/12 h 10 mg/kg e/12 h 10 mg/kg e/8 h 10 mg/kg e/12 h 10 mg/kg e/8 h 10 mg e/6 h	Peak: 20–40 mcg/mL: 65.6% (*n* = 99) Trough: 5–10 mcg/mL: 51% (*n* = 77)	Peak: <20 mcg/mL: 29.1% > 40 mcg/mL: 5.3% Trough: <5 mcg/mL: 32.5% >10 mcg/mL: 16.6%	Nephrotoxicity and ototoxicity from vancomycin in this study are unlikely
McDougal A 1995 [27]	Weight and age (PMA) 1) <0.8 kg + <27 w 2) 0.8–1.2 kg + 27–30 w 3) 1.2–2 kg + 31–36 w 4) >2 kg + ≥37 w	N/A	1)18 mg/kg e/36 h 2)16 mg/kg e/24 h .3)18 mg/kg e/18 h 4)15 mg/kg e/12 h	1) 0% (*n* = 0) 2) P: 62.5% (*n* = 16) T: 18.8% (*n* = 3) 3) P: 73.3% (*n* = 11) T: 20% (*n* = 3) 4) P: 46.2% (*n* = 6) T: 38.5% (*n* = 5) Total: Peak 75%; Trough 25%	1) I: 0%; S: 0% 2) P: I:31.2%; S: 6.3% T: I: 81.3% 3) P: I: 26.7% T: 80% 4) P: I 38.5%; S: 15.4% T: I: 46.2%; S: 15.4%	No adverse effects. No bacteriologic treatment failure.
Patel AD 2013 [28]	Creatinine +/– age (PMA) Inter. Inf: <0.33 mg/dl 0.34–0.44 mg/dl 0.45–0.72 mg/dl 0.73–1.13 mg/dl >1.13 mg/dl Cont. Inf: <0.45 mg/dl + PMA ≥40 w <0.45 mg/dl + PMA <40 w 0.45–0.68 mg/dl + PMA All w >0.68 mg/dl + PMA All w	Inter. inf: No loading dose Cont. inf: 15 mg/kg	Inter. inf: 20 mg/kg e/8 h 15 mg/kg e/8 h 10 mg/kg e/8 h 10 mg/kg e/12 h 15 mg/kg and adjustment Cont. inf: 60 mg/kg/day 50 mg/kg/day 40 mg/kg/day 30 mg/kg/day 20 mg/kg/day	Inter. inf: 10–20 mcg/mL: 46% Cont. inf: Includes 60 mg/kg guideline: 15–25 mcg/mL: 68% (*n* = 41) No 60 mg/kg regimen 15–25 mcg/mL: 82% (*n* = 49)	Inter. inf: <10 mcg/mL: 20% Cont. inf: Includes 60 mg/kg guideline: >25 mcg/mL: 30% <15 mcg/mL: 2% No 60 mg/kg regimen >25 mcg/mL: 5% <15 mcg/mL: 13%	No adverse effects. and no problems with intravenous access.
Plan 2008 [29]	Creatinine 1) ≤ 1.02 mg/dl >1.02 mg/dl 2) ≤ 1.02 mg/dl >1.02 mg/dl	N/A	1) 25 mg/kg/day 15 mg/kg/day 2) 30 mg/kg/day 20 mg/kg/day	1) 10–25 mcg/mL: 74% (*n* = 54) 2) 10–25 mcg/mL: 75% (*n* = 54) Negativization of CoNS* (48 h) = Bacteriological efficacy: 71.3% (*n* = 57/80) 1) 69% (*n* = 27) 2) 73% (*n* = 30) Negativization of blood cultures at the end of treatment: 93% (*n* = 76)	1) <10 mcg/mL: 24% >25 mcg/mL: 1.4% 2) < 10 mcg/mL: 5% >25 mcg/mL: 19% Positivity of CoNS (48 h): 28.7% (*n* = 23/80) Bacteriological inefficacy: 1) 31% (*n* = 12) 2) 27% (*n* = 11) Positivity of blood cultures at the end of treatment: 7% (*n* = 6)	Nephrotoxicity was not evaluated. Creatinine levels were measured at 48 h: similar in both groups: 64 (50–85) mmol/l Vs. 63 (49–85) mmol/l
Demirel 2015 [30]	Age (PMA and PNA) Group 1 PMA ≤29 w + PNA 0–14 d PMA ≤29 w + PNA >14 d PMA 30–36 w + PNA 0–14 d PMA 30–36 w + PNA >14 d PMA 37–44 w + PNA 0–7 d PMA 37–44 w + PNA >7 dPMA ≥ 45 w + PNA All Group 2 PMA ≤29 w + PNA 0–14 d PMA ≤29 w + PNA >14 d PMA 30–36 w + PNA 0–14 d PMA 30–36 w + PNA >14 d PMA 37–44 w + PNA 0–7 d PMA 37–44 w + PNA >7 d PMA ≥ 45 w + PNA All	Group 1: N/A Group 2: 10 mg/kg	Group 1 10 mg/kg e/18 h 10 mg/kg e/12 h 10 mg/kg e/12 h 10 mg/kg e/8 h 10 mg/kg e/12 h 10 mg/kg e/8 h 10 mg e/6 h Group 2: Total daily dose was calculated from the dosage of intermittent administration (cumulative dose)	Clinical failure: Group 1: (–) Group 2: 5.6% (*n* = 2) Tollner score: Group 1: –6 (–7/–4) Group 2: –4.5 (–6/–3) Patients with positive blood cultures at the beginning and became negative at 48 h: Group 1: 57.9% (*n* = 11) Group 2: 63.6% (*n* = 7) Plasma levels: 1) 5–10 mcg/mL: 34.1% (*n* = 14) 2)15–20 mcg/mL: 52.8% (*n* = 19)	Plasma levels: Group 1: a) <5 mcg/mL: 26.8% b) > 10 mcg/mL: 39% Group 2: a) < 15 mcg/mL: 41.7% b) >20 mcg/mL: 5.6%	No adverse effects in any groups. All the infants passed the hearing–screening tests.
Gwee A 2019 [31]	Inter. inf: age (PMA) <29 w 29–35 w 36–44 w >44 w Cont. inf: Cr + age (PMA) <0.45 mg/dl + PMA ≥40 w <0.45 mg/dl + PMA <40 w 0.45–0.68 mg/dl + PMA All w >0.68 mg/dl + PMA All w	Inter. inf: No loading dose Cont. inf: 15 mg/kg	Inter. inf: 15 mg/kg e/24 h 15 mg/kg e/12 h 15 mg/kg e/8 h 15 mg/kg e/6 h Cont. inf: 50 mg/kg/day 40 mg/kg/day 30 mg/kg/day 20 mg/kg/day	Inter. inf: 41,18% (*n* = 21) Cont. inf: 85% (*n* = 45)	Inter. inf: I: 47,06% S: 11,76% Cont. inf: I: 5,67% S: 9,43%	There were no differences in increased creatinine levels or toxicity between groups.

GE: gestational age; PNA: postnatal age; PMA: postmenstrual age; N/A: not applicable; Inter. inf; intermittent infusion; Cont. inf: continuous infusion; T: trough; P: peak; I: infratherapeutic; S: supratherapeutic; NRF: normal renal function; IRF: impaired renal function; P: peak; T: Trough; CoNS: coagulase–negative staphylococci.

**Table 4 antibiotics-10-00347-t004:** Detailed search strategy.

Database	Search Strategy
PubMed	(“vancomycin”[MeSH Terms] OR “vancomycin”[All Fields]) AND (“infant, newborn”[MeSH Terms] OR (“infant”[All Fields] AND “newborn”[All Fields]) OR “newborn infant”[All Fields] OR “neonates”[All Fields]) AND (“pharmacokinetics”[Subheading] OR “pharmacokinetics”[All Fields] OR “pharmacokinetics”[MeSH Terms])
EMBASE	(“vancomycin”/exp OR vancomycin) AND neonates AND (“pharmacokinetics”/exp OR pharmacokinetics)

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
