# Peer review of "Target Attainment and Clinical Efficacy for Vancomycin in Neonates: Systematic Review"

_antibiotics, 2021, doi:10.3390/antibiotics10040347_

Round 1
Reviewer 1 Report
I have reviewed a manuscript entitled Target Attainment and Clinical Efficacy for Vancomycin in Neonates: Systematic Review written by Mejias Trueba et al.
The study is a systematic review with the aim to find the correct therapeutic dosing of Vancomycin by describe the relationship between therapeutic regimens proposed for the neonatal population, and the achievement of clinical or pharmacokinetic objectives, as well as their safety.
The authors have done a bibliographic search starting with 476 articles, and narrowing that down to 20 articles.
I think the manuscript I well written and interesting, but I do have a few comments.
My major concern is that table 3 in its current form is almost impossible to read. Table 3 must be reconstructed and made more easy to read.
I also have a few minor concerns.
Table 1: The Robins and ROB-2 tools need further explanation.
Line 76-77: There is a total of 20 studies, but you mention 16 plus five studies. That adds up to 21 studies.
I think the manuscript can be published after minor revision
Author Response
Attached you will find the corrected version of the manuscript with the modifications, made and indicated with “track changes”, as indicated.
Below are our comments for each of the reviewers. We hope we have resolved all of the reviewers' concerns. However, do not hesitate to contact us if the need arises
1. Table 3 must be reconstructed and made more easy to read.
We fully understand that the tables are difficult to read. We believe that one of the reasons for this may be the change in format that has occurred, so we have tried to restructure it both tables.
The truth is that this point has been the most complex to carry out, since Table 3 exposes the most relevant findings of our study, which makes it difficult to simplify it without losing information of interest.
We have decided to remove the “pharmacokinetic model” column and have better summarized the “safety” section. In addition, we have unified the columns referring to the dosage regimens (PMA, PNA, creatinine ...), creating a new and unique column called "variables involved" to make it easier to read. Finally, we have standardized all the abbreviations and have eliminated the parentheses that indicate the number of patients from the column “Infra / Supratherapeutic”, leaving only the percentages.
Please excuse me that in table 3 the "change track" does not appear correctly, we have had to remove it due to format issues.
- Table 1: The Robins and ROB-2 tools need further explanation.
As indicated, the meaning of both tools (ROBINS and ROB-2) has been explained in greater detail in Table I.
- Line 76-77: There is a total of 20 studies, but you mention 16 plus five studies. That adds up to 21 studies.
Totally agree, this is a mistake that we have corrected: “15 studies were assessed as having some concerns or a moderate risk of bias, and five studies were reported as poor quality or having a serious risk of bias”
Reviewer 2 Report
The Authors extensively described vancomycin use in the neonatal population focusing on the association between the therapeutic regimens and clinical and pharmacokinetic outcomes.
The argument was well developed following an adequate methodology. Moreover, the limitations of the study were described in a suitable manner.
The main findings of this review are: 1) great variability in vancomycin administration regimens, 2) great variability in pharmacokinetic target and their achievement. As a consequence, the Authors can not give specific recommendations except for a probable superiority of continuous infusion over intermittent infusion on the attainment of target concentrations.
However, of the 20 studies analysed, only one is a RCT and only 2 evaluated efficacy so that the main conclusion was that new studies, especially RCTs, are needed. This item represents an important limitation of the paper. In addition to this, as stated by the Authors, the quality of some of the studies included in the review was not excellent because of different kinds of bias.
On the other hand, the paper seems to be quite well written and an adequate methodology was followed.
I suggest:
1) please, organize a "limitations of the study" section. The limitations were described in different parts of the manuscript while it could be preferable to have a specific section for them. Moreover, the delay between articles' research and the publication of the review should be mentioned but as a last limitation.
2) some grammatical and/or English language revisions could be necessary (especially at lines 173, 177, 209-210, 222, 233-237)
3) tables 2 and 3 are very complete and exhaustive but a bit difficult to read. Please try to simplify and make them more readable
Author Response
Attached you will find the corrected version of the manuscript with the modifications, made and indicated with “track changes”, as indicated.
Below are our comments for each of the reviewers. We hope we have resolved all of the reviewers' concerns. However, do not hesitate to contact us if the need arises.
- Please, organize a "limitations of the study" section. The limitations were described in different parts of the manuscript while it could be preferable to have a specific section for them. Moreover, the delay between articles' research and the publication of the review should be mentioned but as a last limitation.
A section on "strengths and limitations" has been created as suggested, and the most important limitations of the study have been grouped, putting in last place the one related to the delay between the bibliographic search and the publication of the article.
- Some grammatical and/or English language revisions could be necessary (especially at lines 173, 177, 209-210, 222, 233-237).
The manuscript has been reviewed again. In addition, we have modified the indicated sentences since, indeed, they had grammatical errors. We hope that they are now resolved.
- Tables 2 and 3 are very complete and exhaustive but a bit difficult to read. Please try to simplify and make them more readable.
We fully understand that the tables are difficult to read. We believe that one of the reasons for this may be the change in format that has occurred, so we have tried to restructure it both tables.
The truth is that this point has been the most complex to carry out, since Table 3 exposes the most relevant findings of our study, which makes it difficult to simplify it without losing information of interest.
We have decided to remove the “pharmacokinetic model” column and have better summarized the “safety” section. In addition, we have unified the columns referring to the dosage regimens (PMA, PNA, creatinine ...), creating a new and unique column called "variables involved" to make it easier to read. Finally, we have standardized all the abbreviations and have eliminated the parentheses that indicate the number of patients from the column “Infra / Supratherapeutic”, leaving only the percentages.
Please excuse me that in table 3 the "change track" does not appear correctly, we have had to remove it due to format issues.